Comprehensive analysis of an lncRNA-miRNA-mRNA competing endogenous RNA network in pulpitis

Lei Fangcao 1
Zhang Han 2
Xie Xiaoli xiexiaoli@csu.edu.cn 1
1 Department of Operative Dentistry and Endodontics, School of Stomatology, Xiangya Stomatological Hospital, Central South University , Changsha , Hunan , People’s Republic of China
2 Institute of Reproductive & Stem Cell Engineering, School of Basic Medical Science, Central South University , Changsha , Hunan , People’s Republic of China
Sotelo-Mundo Rogerio
Electronic publication date: 2019 Jun 17
Publication date: 2019
Volume: 7
Electronic Location ID: e7135
Received 2019 Feb 20; Accepted 2019 May 16
Copyright: ©2019 Lei et al.
Copyright year: 2019
Copyright holder: Lei et al.
License: This is an open access article distributed under the terms of the Creative Commons Attribution License, which permits unrestricted use, distribution, reproduction and adaptation in any medium and for any purpose provided that it is properly attributed. For attribution, the original author(s), title, publication source (PeerJ) and either DOI or URL of the article must be cited.
License URL: https://creativecommons.org/licenses/by/4.0/

Keywords: Pulpitis, Competing endogenous RNA network, Long noncoding RNA, Bioinformatics analysis

Funding: Major Scientific Project of Changsha Science and Technology Bureau kq1804007 Fundamental Research Funds for the Central Universities of Central South University 502211704 This work was supported by the Major Scientific Project of Changsha Science and Technology Bureau (No. kq1804007) to XL Xie, and the Fundamental Research Funds for the Central Universities of Central South University (No.502211704) to FC Lei. The funders had no role in study design, data collection and analysis, decision to publish, or preparation of the manuscript.

==============================
Background

Pulpitis is a common inflammatory disease that affects dental pulp. It is important to understand the molecular signals of inflammation and repair associated with this process. Increasing evidence has revealed that long noncoding RNAs (lncRNAs), via competitively sponging microRNAs (miRNAs), can act as competing endogenous RNAs (ceRNAs) to regulate inflammation and reparative responses. The aim of this study was to elucidate the potential roles of lncRNA, miRNA and messenger RNA (mRNA) ceRNA networks in pulpitis tissues compared to normal control tissues.

Methods

The oligo and limma packages were used to identify differentially expressed lncRNAs and mRNAs (DElncRNAs and DEmRNAs, respectively) based on expression profiles in two datasets, GSE92681 and GSE77459, from the Gene Expression Omnibus (GEO) database. Differentially expressed genes (DEGs) were further analyzed by Gene Ontology (GO) and Kyoto Encyclopedia of Genes and Genomes (KEGG) pathway enrichment analyses. Protein–protein interaction (PPI) networks and modules were established to screen hub genes using the Search Tool for the Retrieval of Interacting Genes/Proteins (STRING) and the Molecular Complex Detection (MCODE) plugin for Cytoscape, respectively. Furthermore, an lncRNA-miRNA-mRNA-hub genes regulatory network was constructed to investigate mechanisms related to the progression and prognosis of pulpitis. Then, quantitative real-time polymerase chain reaction (qRT-PCR) was applied to verify critical lncRNAs that may significantly affect the pathogenesis in inflamed and normal human dental pulp.

Results

A total of 644 upregulated and 264 downregulated differentially expressed genes (DEGs) in pulpitis samples were identified from the GSE77459 dataset, while 8 up- and 19 downregulated probes associated with lncRNA were identified from the GSE92681 dataset. Protein–protein interaction (PPI) based on STRING analysis revealed a network of DEGs containing 4,929 edges and 623 nodes. Upon combined analysis of the constructed PPI network and the MCODE results, 10 hub genes, including IL6, IL8, PTPRC, IL1B, TLR2, ITGAM, CCL2, PIK3CG, ICAM1, and PIK3CD, were detected in the network. Next, a ceRNA regulatory relationship consisting of one lncRNA (PVT1), one miRNA (hsa-miR-455-5p) and two mRNAs (SOCS3 and PLXNC1) was established. Then, we constructed the network in which the regulatory relationship between ceRNA and hub genes was summarized. Finally, our qRT-PCR results confirmed significantly higher levels of PVT1 transcript in inflamed pulp than in normal pulp tissues (p = 0.03).

Conclusion

Our study identified a novel lncRNA-mediated ceRNA regulatory mechanisms in the pathogenesis of pulpitis.

Introduction

Inflammation of the dental pulp (pulpitis) is associated with microbial infection of the root canal system and the host response, which is characterized by spontaneous or provoked pain (Bender, 2000; Rocas et al., 2015). Acute pulpitis can be an extremely painful condition and is believed to be one of the main reasons patients seek emergency dental treatment (Currie et al., 2017). Depending on the state of inflammation, various treatment regimens are currently applied clinically, including nonsurgical root canal treatment (such as pulpotomy and pulpectomy) and surgical endodontic treatment (European Society of Endodontology, 2006). If the issue cannot be solved in time or appropriately, pulpitis may progress to pulp necrosis, periapical periodontitis, or even severe oral and maxillofacial space infections (Bertossi et al., 2017), ultimately causing a significant medical and economic burden in terms of treatment costs.

Pulp inflammation represents a complex physiological response to harmful stimuli, such as bacterial infections and physical and chemical injuries (Yu & Abbott, 2016; Larsen & Fiehn, 2017; Pedano et al., 2018). The balance between inflammation and reparative processes in host defense reactions can determine the extent of pulp inflammation, in which multiple signaling pathways are involved (Cooper, Holder & Smith, 2014; Farges et al., 2015). Studies over the past decade have provided much information on the role of epigenetic modifications in inflammatory diseases such as pulpitis and periodontitis. These studies have investigated modifications of histones, methylation of DNA, and regulation of noncoding RNAs (ncRNAs), among other epigenetic mechanisms (Kearney et al., 2018).

Due to the application of sequencing and bioinformatics approaches, recent evidence has identified previously unannotated transcripts and ncRNAs involved in pathologic mechanisms, such as microRNAs (miRNAs), long noncoding RNAs (lncRNAs), pseudogenes and circular RNAs, which were previously believed to have no biological functions (Kaikkonen & Adelman, 2018). MiRNAs are small (∼21–24 nucleotides in length), single-stranded ncRNA molecules that can bind to complementary sequences within the 3′ untranslated regions (UTRs) of target messenger RNAs (mRNAs), resulting in mRNA degradation or repression (Afonso-Grunz & Muller, 2015). LncRNAs, which are ncRNAs of >200 nucleotides, function as competing endogenous RNAs (ceRNAs) that interact with mRNAs, serving as miRNA sponges to restrain miRNA function by competing for miRNA response elements (MREs) (Tay, Rinn & Pandolfi, 2014). Increasing evidence has illustrated the essential roles of differentially expressed miRNAs and lncRNAs (DEmiRNAs and DElncRNAs, respectively) in a variety of cellular and pathologic processes associated with pulpitis, in which abnormal expression occurs (Zhong et al., 2012; Huang & Chen, 2018). However, there have been few comprehensive analyses of pulpitis-associated lncRNAs and miRNAs in the context of a ceRNA network. The discovery of ceRNA network interactions could provide important insights to advance our understanding of the pathogenesis of pulpitis.

In this work, we first performed an integrated analysis and identified differentially expressed genes (DEGs) between pulpitis and matched normal pulp tissues from two gene expression profiles in the Gene Expression Omnibus (GEO) database. Then, Gene Ontology (GO) enrichment and Kyoto Encyclopedia of Genes and Genomes (KEGG) pathway analysis were further conducted to analyze the major biological functions of the DEGs. Next, we examined potential crosstalk in a constructed lncRNA-miRNA-mRNA ceRNA network that involved one core lncRNA, one miRNA, and two mRNAs. Moreover, ten hub genes related to pulpitis were identified as being aberrantly expressed in the protein-protein interaction (PPI) network. Ultimately, the DElncRNA plasmacytoma variant translocation gene 1(PVT1), which was significantly upregulated in inflamed pulp tissue, was validated by quantitative real-time polymerase chain reaction (qRT-PCR).To the best of our knowledge, this study is the first attempt to apply bioinformatics approaches to investigate the differential expression profiles of a specific lncRNA-mediated ceRNA network in inflamed pulp.

Materials & Methods

Microarray data

Microarray data from the GSE92681 and GSE77459 datasets were downloaded from the GEO (https://www.ncbi.nlm.nih.gov/geo) of NCBI. GSE92681 including seven pulpitis and five matched normal pulp tissues was associated with the GPL16956 platform (Agilent-045997 Arraystar Human LncRNA Microarray V3) (Huang & Chen, 2018). GSE77459 contained 6 pulpitis tissue specimens and six matched normal samples, and was associated the GPL17692 platform (Affymetrix Human Gene 2.1 ST Array) (Galicia et al., 2016). The DEmiRNAs were retrieved as reported by Zhong et al. (2012); until now, that data was the only known DEmiRNAs between normal and inflamed human pulp tissue. All data were freely accessible online.

Data processing

The oligo package was used to read the microarray and normalize the expression data (Carvalho & Irizarry, 2010). After the raw data were transformed to the expression files, the expression files were further processed with the Linear Models for Microarray data (limma) R/Bioconductor package (http://www.bioconductor.org) for analysis of the DEGs between pulpitis samples and control samples with biological replication; the Benjamin and Hochberg method was applied for multiple testing corrections (Ritchie et al., 2015). Genes that met the cutoff criteria, a corrected p-value < 0.05 and a —log2 fold change (FC)— > 1, were considered DEGs.

Function and pathway enrichment analysis of the DEmRNAs

To investigate pulpitis progression at the functional level, GO and pathway enrichment analyses of the identified DEmRNAs were performed in this study. The identified DEGs were enriched for terms in the GO biological process (BP), molecular function (MF), and cellular component (CC) categories (Ashburner et al., 2000). KEGG analysis was utilized to interpret the potential functions and pathways of the aberrantly expressed genes (Kanehisa et al., 2010). In addition, the above data were analyzed using the clusterProfiler package in R (Yu et al., 2012).

Protein–Protein Interaction (PPI) network analysis

The Search Tool for the Retrieval of Interacting Genes/Proteins (STRING, https://string-db.org/) was used to identify PPIs among the selected DEGs. A combined score ≥0.4 was chosen for PPI network construction (Szklarczyk et al., 2015). Cytoscape software was used to visualize the established DEGs network, while the MCODE plugin and CentiScaPe in Cytoscape were used to select significant modules and calculate the connective of genes from the PPI network (Shannon et al., 2003; Scardoni et al., 2014).

Construction of the ceRNA network

The lncRNA-associated ceRNA network in pulpitis was next constructed. First, miRNA-lncRNA interactions were evaluated using the starBase (version 2.0) database with the default parameters (clade: mammal, genome: human, assembly: hg19, number of supporting experiments: ≥1) (Li et al., 2014). Next, starBase (version 2.0), TargetScan (release 7.1), and miRDB (last modified: May 03, 2016) were used to retrieve miRNA-targeted mRNAs (Li et al., 2014; Agarwal et al., 2015; Wong & Wang, 2015). The main steps of the method are shown in Fig. 1.

Figure 1 Main steps of the construction of the regulatory network in pulpitis.

Step 1: We identified the differentially expressed mRNAs, lncRNAs and miRNAs. Step 2: The GO and KEGG enrichment analyses were conducted and the PPI network was constructed. Then, we identified the hub genes from the PPI network. Step 3: The ceRNA regulatory relationships were predicted using online tools. Step 4: The ceRNA network with hub genes was constructed.

Validation based on clinical samples of human dental pulp

To further verify the expression of the DElncRNAs, qRT-PCR was performed. Briefly, four human inflamed pulp tissue samples and four normal pulp tissue samples were obtained from Xiangya Stomatological Hospital. The normal pulp tissues were collected from healthy third molars extracted for orthodontic purposes. The inflamed pulp tissues were extracted from teeth diagnosed with irreversible pulpitis in accordance with the endodontic diagnosis system from the American Association of Endodontists. All patients gave written informed consent. The Ethics Committee of Xiangya Stomatological Hospital of Central South University granted Ethical approval to carry out the study within its facilities (Ethical Approval number: 20180026). Next, RNA from all tissue samples, which were previously preserved in liquid nitrogen, was extracted with TRIzol reagent (Thermo Fisher, Waltham, MA, USA). Reverse transcription was performed with a Transcriptor First Strand cDNA Synthesis Kit (Roche, Indianapolis, IN, USA). qRT-PCR was used to monitor the expression of PVT1 (Fw, 5′-TGAGAACTGTCCTTACGTGACC-3′; Rev, 5′-AGAGCACCAAGACTGGCTCT-3′) and GAPDH (Fw, 5′-GACAGTCAGCCGCATCTTCTT-3′; Rev, 5′-AATCCGTTGACTCCGAC CTTC-3′); GAPDH was used as the housekeeping gene for normalization. Amplification was performed with LightCycler 480 SYBR Green I Master (Roche, Indianapolis, IN, USA) on a real-time PCR system (LightCycler 480; Roche, Indianapolis, IN, USA). Threshold cycle values (CT) were determined, and the data were analyzed with Roche software with the 2−ΔΔCT method. An unpaired Wilcoxon test was conducted to compare lncRNA expression at the transcriptional level between normal and inflamed pulp tissues.

Results

Identification of DEGs in pulpitis

Based on the criteria of p < 0.05 and —log2 FC— > 1, a total of 908 DEGs were screened from GSE77459, including 264 downregulated genes and 644 upregulated genes in pulpitis. In the GSE92681 dataset, 27 probes associated with DElncRNA, including 19 downregulated and eight upregulated probes, were identified. A volcano plot for GSE77459 and a heatmap for GSE92681 are shown in Fig. 2. The detailed differential expression profiles are summarized in File S1.

Figure 2 DEGs between pulpits samples and normal samples.

(A) Volcano plot for the DEGs in dataset GSE77459. The x-axis indicates the log FC, and the y-axis indicates the log10 (adjusted p-value). The red dots represent upregulated genes, and the blue dots represent downregulated genes. The DEGs were screened on the basis of a |fold change| > 1.0 and an adjusted p-value of < 0.05. The black dots represent genes with no significant difference. FC, fold change. (B) Heatmap of the DEGs in dataset GSE92681. The relative expression values were normalized to fall in a range from zero to one. Genes expressed at high levels are shown in blue, while those expressed at low levels are shown in white.

GO and KEGG analysis of the DEGs

GO function and KEGG pathway enrichment analyses of the DEGs were performed using the clusterProfiler package in R. The results of GO analysis illustrated that the DEGs were enriched for BP terms including leukocyte migration, the adaptive immune response, the immune response-regulating cell surface receptor signaling pathway, regulation of leukocyte activation, and the immune response-activating cell surface receptor signaling pathway. CC analysis showed that the DEGs were significantly enriched for the side of membrane, secretory granule membrane, and external side of plasma membrane, tertiary granule, and specific granule terms. For the MF category, the DEGs were enriched in antigen binding, serine-type peptidase activity, serine hydrolase activity, serine-type endopeptidase activity, and cytokine activity (Fig. 3).

Figure 3 Top 10 processes revealed in GO enrichment to influence biological process (BP), molecular function (MF), and cellular component (CC).

The colored dots represent the term enrichment: blue indicates low enrichment, and red indicates high enrichment. The sizes of the dots represent the numbers of genes in each GO category.

Based on KEGG pathway analysis, the DEGs were significantly enriched in pathways associated with cytokine-cytokine receptor interaction, the chemokine signaling pathway, cell adhesion molecules, Staphylococcus aureus infection, and the hematopoietic cell lineage (Fig. 4). The detailed results of the GO enrichment and KEGG pathway analyses are provided in Supplementary File 2.

Figure 4 Top 10 enriched KEGG pathways for the DEGs.

The y-axis shows the KEGG pathway names. The colored dots represent the term enrichment: blue indicates low enrichment, while red indicates high enrichment. The sizes of the dots represent the numbers of genes.

PPI network construction and hub genes identification

Protein interactions among the DEGs were detected with the online STRING program with a cutoff score of ≥0.4. In total, 4,929 edges and 623 nodes were involved in the PPI network. One significant module containing 528 edges and 33 nodes was selected from the PPI network by the MCODE plugin in Cytoscape (Fig. 5). In this module, we found that most of the genes were mainly enriched for and associated with chemokines (Tables 1 and 2).

Figure 5 The most significant module in the PPI network of the DEGs.

The most significant module includes 528 edges and 33 nodes. The circles represent genes, and the lines represent interactions between the proteins encoded by the genes.

Table 1 Top 15 enriched GO terms of the DEGs.

Terms	Pathway description	Count	P-Value	
GO.BP:0070098	chemokine-mediated signaling pathway	17	4.46E-35	
GO.BP:0060326	cell chemotaxis	21	2.18E-33	
GO.BP:0002690	positive regulation of leukocyte chemotaxis	17	2.84E-30	
GO.BP:0050921	positive regulation of chemotaxis	18	7.66E-30	
GO.BP:0006954	inflammatory response	23	5.00E-29	
GO.CC:0005615	extracellular space	15	1.79E-07	
GO.CC:0009897	external side of plasma membrane	6	0.000948	
GO.CC:0005886	plasma membrane	19	0.00174	
GO.CC:0071944	cell periphery	19	0.00174	
GO.CC:0005887	integral component of plasma membrane	10	0.0144	
GO.MF:0045236	CXCR chemokine receptor binding	11	5.28E-25	
GO.MF:0008009	chemokine activity	13	2.39E-24	
GO.MF:0001664	G-protein coupled receptor binding	15	8.21E-20	
GO.MF:0048248	CXCR3 chemokine receptor binding	5	4.67E-12	
GO.MF:0008528	G-protein coupled peptide receptor activity	9	2.18E-11	

Table 2 Top five enriched KEGG pathways of the DEGs.

Category	Pathway description	Count	P-Value	
KEGG:map04062	chemokine signaling pathway	21	1.49E-33	
KEGG:map04060	cytokine-cytokine receptor interaction	20	4.10E-28	
KEGG:map05150	Staphylococcus aureus infection	6	1.78E-08	
KEGG:map04080	neuroactive ligand–receptor interaction	9	2.16E-08	
KEGG:map04668	TNF signaling pathway	6	9.52E-07	

In addition, we screened the top 10 hub mRNAs from the PPI network. These main hub genes were IL6, IL8, PTPRC, IL1B, TLR2, ITGAM, CCL2, PIK3CG, ICAM1, and PIK3CD. Hence, the key genes associated with pulpitis could be predicted by our network.

Construction and analysis of the ceRNA network in Pulpitis

To identify correlations among the DElncRNAs in the ceRNA network, we used starBase to search for interactions between miRNAs and lncRNAs. The target mRNAs of the miRNAs in the network were predicted with starBase, TargetScan, and miRDB. The ceRNA network was constructed based on coexpressed lncRNAs/miRNAs, miRNAs/mRNAs, and lncRNAs/mRNAs. Overlapping datasets were visualized using Venn diagrams (Fig. 6A). As shown in Fig. 6B, the lncRNA-miRNA-mRNA network was composed of one lncRNA (PVT1), one miRNA (hsa-miR-455-5p), and two mRNAs (suppressor of cytokine signaling 3, SOCS3 and Plexin C1, PLXNC1). In addition, a comprehensive analysis of the relationships among PVT1, hsa-miR-455-5p, SOCS3 and PLXNC1 and the top 10 downstream connected genes is depicted in Fig. 6B.

Figure 6 Construction of ceRNA network in pulpitis.

(A) Venn diagram showing the number of distinct and overlapping RNAs among the upregulated genes and the RNAs identified with miRDB, starBase, and TargetScan. The overlapping areas show the upregulated genes identified by three online tools. (B) Interaction of RNAs in the PVT1-associated ceRNA network. The triangle node and the diamond node represent the lncRNA and miRNA, respectively. The rectangle nodes represent miR-455-5p-targeted mRNAs. The round nodes are the top 10 hub DEmRNAs in the network. The up- and downregulated genes are colored in red and green, respectively.

Validation of PVT1 expression in clinical samples of human dental pulp

We evaluated the expression of PVT1 by qRT-PCR in pulpitis tissues compared to normal pulp tissues. As depicted in Fig. 7, PVT1 transcript levels were significantly higher in inflamed pulp than in normal pulp (p < 0.05).

Figure 7 Relative expression levels of PVT1 in inflamed and normal pulp tissue.

The transcript levels of PVT1 were determined by qRT-PCR and normalized to those of the reference RNA GAPDH. p-value = 0.03.

Discussion

Pulpitis is considered a tightly regulated process involving microorganisms and host immune events mediated by molecular factors (Renard et al., 2016). Inflammation of the dental pulp is characterized by opportunistic infection of the pulp space by commensal oral microorganisms, such as Porphyromonas and Streptococcus species (Rocas et al., 2016). The most common route for microorganism invasion is through dental caries. Other possible portals include tooth damage from trauma, exposed dentinal tubules or the apical foramen (Raslan & Wetzel, 2006). Upon irritation, cells in human dental pulp, for example, endothelial cells, odontoblasts and macrophages, immediately trigger immune responses to pathogens and their virulent factors, potentially stalling the spread of infection, preventing injury-related signaling, and launching reparative processes (Rechenberg, Galicia & Peters, 2016). However, if the delicate balance between the immune-inflammatory response and dental tissue healing is disrupted, irreversible pulpitis could result from uncontrollable inflammation caused by the invading bacteria.

Over the past decades, ncRNAs, including lncRNAs and short ncRNAs (such as miRNAs), have gained increasing attention for their roles in physiological and pathologic responses. In diverse biological processes, gene expression is regulated by posttranscriptional mechanisms involving ncRNAs through their binding to the 3′-UTRs of target mRNAs, leading to translational repression or target degradation (Vasudevan, 2012). Accumulating studies have provided evidence supporting the ceRNA hypothesis, which holds that lncRNAs harboring MREs can competitively bind to certain miRNAs, thus regulating miRNA-mediated downstream target gene silencing at the posttranscriptional level (Khorkova, Hsiao & Wahlestedt, 2015). Although the functional relevance of several lncRNAs and miRNAs in pulpitis has been proven in the scientific literature, an lncRNA-based ceRNA network involved in pulpitis has yet to be defined (Zhong et al., 2012; Huang & Chen, 2018). In the current study, bioinformatics analyses were applied to explore the expression profiles of lncRNAs, miRNAs, and mRNAs in patients with pulpitis. Furthermore, an lncRNA-related ceRNA network was constructed by integrating data from GEO pulpitis expression profiles. These data might shed light on how lncRNA-miRNA-mRNA interactions are involved in pulpitis. Immune response can be triggered by inflammatory mediators or cytokines produced by infectious cells through the transcriptional or post-transcriptional regulations. The expression of ncRNAs may be closely related to the development of immune cells (particularly monocytes, macrophages, NK cells, T helper cells, CD8+ T cells, and Treg cells) and the maintenance of immune system homeostasis (Cortez et al., 2019; Imamura & Akimitsu, 2014). For instance, lincRNA-Cox2 regulates the release of inflammatory mediators and cytokines by stimulation of TLR2 in bacterial infections (Carpenter et al., 2013), and miR-155 exerts an antiproliferative effect on CD8+ T cells in response to the type I interferon signaling (Gracias et al., 2013).

LncRNAs and miRNAs, which are closely correlated with inflammatory mediator production and pulp repair, could be affected by the inflammatory status in dental pulp (Zhong et al., 2012; Galicia et al., 2016; Huang & Chen, 2018). In this study, our findings indicated that lncRNA PVT1might regulate the progression of pulpitis by acting as a sponge for miR-455-5p. Emerging studies have proven that PVT1 plays an important role in inflammation and tumorigenesis. For example, down-regulation of PVT1 correlates with the differentiation of Th17 cells and the duration of multiple sclerosis, a chronic immune-mediated disease (Eftekharian et al., 2017). Additionally, PVT1 promotes the production of inflammatory cytokines to aggravate the progression of IL-1β-stimulated osteoarthritis (Zhao et al., 2018). Overexpressed PVT1 can enhance the expression of IL-6 and IL-1β in a nonbinding manner by regulating the nuclear factor-κB (NF-κB) pathway, which is considered a critical mechanism in the regulation of immune and inflammatory processes, and ultimately aggravating septic acute kidney injury (Huang et al., 2017). Moreover, some studies have revealed that PVT1, which is located on 8q24.21 and contains the myc proto-oncogene, is upregulated in certain human tumors (Shtivelman & Bishop, 1989) and is correlated with pathologic stage in multiple cancers (Chai et al., 2018). In this study, our qRT-PCR results indicated that PVT1 was upregulated in pulpitis tissue compared with matched normal pulp tissue, suggesting that PVT1 is an important factor contributing to pulpitis. However, the exact role of PVT1 in the occurrence and development of pulp inflammation and the underlying mechanism require further investigation.

MiR-455-5p is one subtype of the mature miR-455, which is located in the sense strand of chromosome 9q32, and is a tumor-associated miRNA molecule. The other subtype is miR-455-3p (Arai et al., 2019). MiR-455 is involved in a variety of biological processes through the repression of specific mRNA at the posttranscriptional level. For example, miR-455–5p regulates the posttranscriptional repression of UDP-glucuronosyltransferase (UGT) 2B expression by binding to MREs in 3′-UTR of UGT2B, which plays a role in drug glucuronidation (Papageorgiou & Court, 2017). Moreover, in colorectal cancer, miR-455 is up-regulated to inhibit the protein expression of RAF proto-oncogene serine/threonine protein kinase (RAF1) other than affects mRNA level, which regulates the cellular proliferation and invasion (Chai et al., 2015). In pulpitis, the precise posttranscriptional mechanisms of miR-455-5p remain to be explored. Previous studies have revealed that miR-455-5p can serve as a prognostic biomarker and therapeutic target for patients with certain cancers, such as non-small cell lung cancer and basal cell carcinoma (Sand et al., 2012; Wang et al., 2017). Moreover, miR-455 represents novel regulator of the immune and inflammatory response.MiR-455-5p plays an anti-inflammatory role in multiple sclerosis. MiR-455-5p is negatively correlated with the activation of the NF-κB pathway, thus inhibiting the expression of IL-1β, IL-6 and IL-8 and in turn ameliorating the severity of multiple sclerosis (Torabi et al., 2019). Macrophage polarization is an essential component of immunity and homeostasis, and miR-455-5p overexpression reverses the polarization of macrophages to the M1 phenotype to reduce the secretion of proinflammatory cytokines, such as IL-1β and TNF-α (Chi et al., 2018). The accumulation of miR-455 regulates the innate immune response in E2F1-deficient mice, leading to reduced inflammation induced by LPS (Warg et al., 2012). In patients infected with the hepatitis B virus, serum miR-455-3p levels are decreased during chronic disease progression (Singh et al., 2018). The algorithms in our study predicted that miR-455-5p was downregulated (Fig. 6B) in pulpitis samples compared to control samples. Further work will be needed with multiple clinical samples to clarify the action of miR-455-5p in pulpitis and thus to uncover the mechanisms underlying the effects of miR-455-5p on proinflammatory processes.

Based on the module produced by the STRING analysis, two core mRNAs, identified as SOCS3 and PLXNC1, can be considered important target genes of miR-455-5p. SOCS3 is a major suppressor of inflammation and is known as a feedback inhibitor of the JAK/STAT3 signaling pathway (Carow & Rottenberg, 2014). SOCS3 can inhibit JAK2 activity to attenuate STAT3 phosphorylation, inhibit the NF-κB pathway and trigger the expression of various genes in response to cytokines (the IL-6 family and IL-10), consequently affecting cell proliferation, differentiation, and apoptosis (Gao et al., 2018). SOCS3 deficiency results in increased alveolar loss with high levels of IL-1β, IL-6, and IL-8 in periodontitis (Papathanasiou et al., 2016). Furthermore, induction of SOCS3 expression may markedly reduce cell adherence by inhibiting TNF-α-stimulated ICAM1 expression in lung inflammation (Lee et al., 2013). PLXNC1 is an endogenous receptor of the neuronal guidance protein semaphorin 7a (Sema 7a). Previous studies have suggested that PLXNC1 might play an important role in immunological and proinflammatory responses. A study by Konig et al. (2014) on PLXNC1−∕− mice provided evidence that PLXNC1 depletion results in a reduced inflammatory response and decreased cytokine IL-6 production in vivo. We hypothesize that PLXNC1 and SOCS3 play important roles in conditions associated with inflammation. Both might therefore be potential targets to prevent damage due to excessive tissue inflammation.

GO and pathway analyses were used to infer the potential functions of the DEGs in pulpitis. GO annotation revealed that the top GO terms in the BP and MF categories for the DEGs between the inflamed and healthy pulp tissue were mainly associated with the immune/inflammation system (Fig. 3), suggesting that the cascade of cellular events in pulpitis is precisely regulated by migrating leukocytes and their surrounding microenvironments and also supporting other previous theories explaining the occurrence of pulpitis (Cooper et al., 2017). Furthermore, KEGG pathway analysis indicated that the cytokine-cytokine receptor interaction pathway was the most highly enriched pathway between pulpitis and normal dental tissues, while other classic inflammatory signaling pathways, including the chemokine signaling pathway, cell adhesion molecules (CAMs), Staphylococcus aureus infection, the hematopoietic cell lineage, the TNF signaling pathway, and osteoclast differentiation were also enriched. These pathways were mainly involved in immune regulation, intercellular signaling, defense mechanisms, osteoclast differentiation, and hematopoiesis.

Conclusions

To the best of our knowledge, there have been no studies on lncRNA-associated ceRNA networks in pulpitis. Here, we used bioinformatics methods to elucidate the lncRNA-miRNA-mRNA ceRNA network associated with dental pulp inflammation. Our analysis suggests a potential mechanism by which PVT1 competes with miR-455-5p to regulate the expression of SOCS3 and PLXNC1. Upregulation of PVT1, which has been validated in inflamed pulp by qRT-PCR, could increase inflammation and cytokine and chemokine production in inflamed pulp tissue compared with healthy pulp tissue. In addition, overexpression of PVT1 reduced miR-455-5p expression and indirectly promoted SOCS3 and PLXNC1 expression and the cytokine cascade. In addition, SOCS3 and PLXNC1 could regulate the hub genes (including IL-1β, IL-6, IL-8 and ICAM1), which play a critical role in pulp inflammation. Upon comprehensive analysis of the lncRNA-related ceRNA network, some novel and crucial characteristics of pulpitis were revealed. These findings provide new insights into the pathogenesis of endodontic lesions and might identify potential diagnostic and therapeutic strategies for future studies.

Supplemental Information

Supplemental Information 1 The detailed differential expression profiles

Click here for additional data file.

Supplemental Information 2 The detailed results of the GO enrichment and KEGG pathway analyses

Click here for additional data file.

Supplemental Information 3 Raw Ct values from qRT-PCR

Click here for additional data file.

Additional Information and Declarations

Competing Interests

Author Contributions

Human Ethics

Data Availability

The authors declare there are no competing interests.

Fangcao Lei and Han Zhang performed the experiments, analyzed the data, contributed reagents/materials/analysis tools, prepared figures and/or tables.

Xiaoli Xie conceived and designed the experiments, contributed reagents/materials/analysis tools, authored or reviewed drafts of the paper, approved the final draft.

The following information was supplied relating to ethical approvals (i.e., approving body and any reference numbers):

The Ethics Committee of Xiangya Stomatological Hospital of Central South University granted Ethical approval to carry out the study within its facilities (Ethical Approval number: 20180026).

The following information was supplied regarding data availability:

The raw data are available in the Supplemental Files.

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
