# Peer review of "Comprehensive analysis of an lncRNA-miRNA-mRNA competing endogenous RNA network in pulpitis"

_PeerJ, doi:10.7717/peerj.7135_

## Round 0.1 · original submission · Major Revisions

Please submit a detailed point-by-point answer to the reviewers, in particular abot the novelty of the work besides donde in pulp tissue. An important point is to elaborate on the role of lnRNA in immune response. One of the reviewers is not convinced of the presence of a hub gene in the network, please elaborate in detail. I concur that the manuscript is well written and and very novel, so I encourage to present a revised version of your contribution.

For ease of review, please include a version of the revised manuscript with track changes to identify the modifications on the document.

·

Basic reporting

1. I suggest detailing further on non-coding RNAs modulating immune responses via post-transcriptional regulation. For example, miR-146a and miR-155 work in a feedback manner to maintain immunological homeostasis. It would be a plus if the authors could relate the state of the art of postranscriptional regulation of microRNAs, specifying on miR-455. This would round up the whole idea introductory idea.

2. A more detailed figure description and including labels from the main steps on Figure 1, as in “Step 1, Step 2... etc.”, can make it easier to the reader to analyze the diagram without recurring to reading the whole document.

3. Figure 2: On Figure 2-A, adding a label that read “Regulation:” before the symbol list from “Down, Not, Up” could help for better understanding by only looking at the figure.

4. Name on Figure 3 seems misleading.
It reads "Top 10 processes revealed in GO enrichment analysis of clusters of the upregulated DEGs." but it apparently show top 10 processes of 3 categories, so maybe stating it as "Top 10 processes revealed in GO enrichment to influence biological process (BP), molecular function (MF), and cellular component (CC).” could guide readers to better analyze the information.

6. Line 359 Has a red mark on a reference

Experimental design

No comment

Validity of the findings

The study and its findings are very interesting. I advise the authors to describe a little bit further about the relation of noncoding RNAs in immunological modulation. This will nicely round up your discussion and conclusions.

Reviewer 2 ·

Basic reporting

1.The manuscript is clearly written in professional english.
2. The format of the manuscript is meet the standard of Peer J. They provided sufficient background and literates.
3.Figures quality is OK.

Experimental design

1.The experiment design is only based on bio-informatics analysis. It is strongly suggested to add some experimental data to validate your conclusion.
2. The original research meets the Aims and Scope of the journal.
3.Research question is well defined and meaningful. However, it is hard to say whether the conclusion of the manuscript is right or not and whether the hypothesis fills an identified knowledge gap.
4. Methods provide the enough information to replicate.
5.Routine technology is used for the investigate.

Validity of the findings

It is very hard to evaluate the novelty of the manuscript now. The conclusion is not very convincing. The authors selected 10 hub genes including IL6, IL8, PTPRC, IL1B, TLR2, ITGAM, CCL2, PIK3CG, ICAM1, and PIK3CD. However, they built a ceRNA network is In the PVT1-hsa-miR-455-5p-SOCS3 and PLXNC1. There is no hub gene at all. In addition, it is strongly suggested the authors do some experiments to evaluate your conclusion in pulptitis tissues and dental pulp cells.

Additional comments

In Figure 2 B, please replace “code name” by real gene name.

---

## Round 0.2 · accepted · Accept

The authors have addressed all the questions from the reviewers and have improved the final version.

·

Basic reporting

Redaction is unambiguous and professional. Correction from authors on previous comments has brought the previously needed references and background to support their statements and findings.

Experimental design

The experimental design was improved with rtqPCR validation.

I agree with authors that in vitro establishment of an in vitro pulpitis model could differ highly from in vivo sampling since there are so many variables involved in the process it is impossible to model it accurately enough.

This study describes with sufficient detail the bioinformatic approach to analyze this inflammatory state and could serve as a platform to develop future studies involving cell culture and specific gene validations of the proposed regulatory hub.

Validity of the findings

Conclusions are well stated, highly improved discussion and conclusions with corrections from reviewers comments.

The information contained in this article will benefit literature and hopefully encourage the scientific community to continue the study of regulation networks relating to non-coding RNAs in different inflammatory states, this is helpful to design and develop more effective therapies.